# Can We Partially Rewrite Transformers in Natural Language?

## Abstract

The greatest ambition of mechanistic interpretability is to completely rewrite deep neural networks in a format that is more amenable to human understanding, while preserving their behavior and performance. In this paper we evaluate whether sparse autoencoders (SAEs) and transcoders can be used for this purpose. We use an automated pipeline to generate explanations for each of the sparse coder latents. We then simulate the activation of each latent on a number of different inputs using an LLM prompted with the explanation we generated in the previous step, and "partially rewrite" the original model by patching the simulated activations into its forward pass. We find that current sparse coding techniques and automated interpretability pipelines are not up to the task of rewriting even a single layer of a transformer: the model is severely degraded by patching in the simulated activations. We believe this approach is the most thorough way to assess the quality of SAEs and transcoders, despite its high computational cost.

## 1 Introduction

While large language models (LLMs) have reached human level performance in many areas [Guo et al., 2025], we understand little about their internal representations. Early mechanistic interpretability research attempted to explain the activation patterns of individual neurons [Olah et al., 2020, Gurnee et al., 2023, 2024], but research has found that most neurons are "polysemantic", activating in semantically diverse contexts [Arora et al., 2018, Elhage et al., 2022].

Sparse autoencoders (SAEs) were proposed to address polysemanticity [Cunningham et al., 2023]. SAEs consist of two parts: an encoder that transforms activation vectors into a sparse, higher-dimensional latent space, and a decoder that projects the latents back into the original space. Both parts are trained jointly to minimize reconstruction error. Recently, a significant effort was made to scale SAE training to larger models, like GPT-4 [Gao et al., 2024] and Claude 3 Sonnet [Templeton et al., 2024], and they have become an important interpretability tool for LLMs. Recently Dunefsky et al. [2024] proposed sparse *transcoders* as an alternative method for extracting interpretable features from LLMs. The architecture of the transcoder is identical to that of an SAE, but it is trained to predict the output of a feedforward network given its input. With a good enough transcoder, we should be able to entirely replace the original FFN with its transcoder approximation, thereby partially rewriting the model in terms of more interpretable primitives.

While it seems that neurons and sparse autoencoder latents can be explained by looking at the examples they activate on [Bills et al., 2023], some are more easily understood by their downstream effects [Gur-Arieh et al., 2025]. Paulo et al. [2024] took inspiration on Bills et al. [2023] and built an automated pipeline for generating natural language explanations of SAE latents and evaluating how good these explanations are. Rigorously measuring how interpretable an explanation is still a complicated and methodologically fraught task.

Submitted to 39th Conference on Neural Information Processing Systems (NeurIPS 2025). Do not distribute.

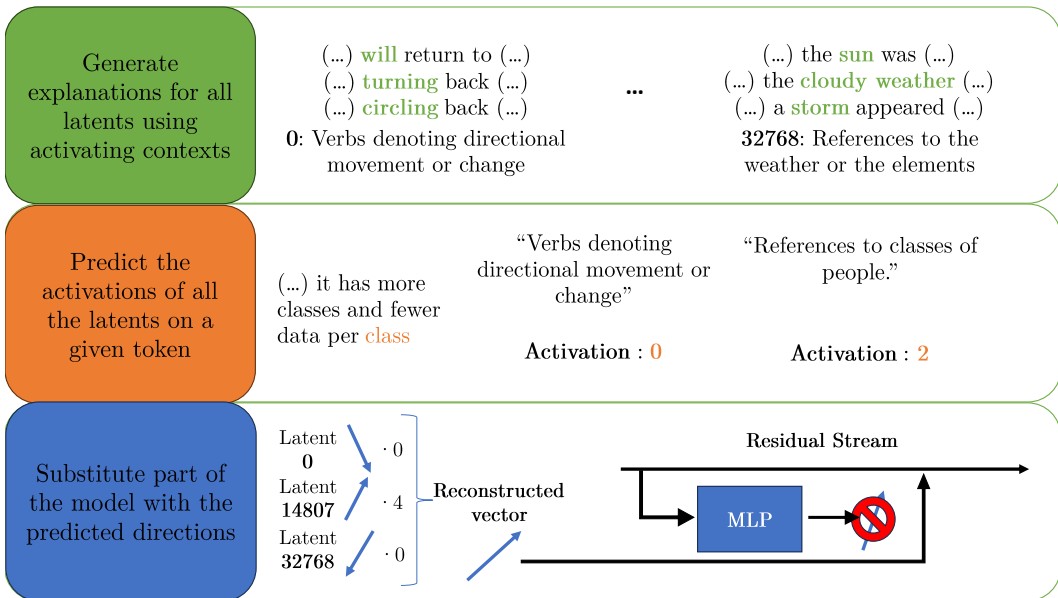

Figure 1: **Partially rewriting an LLM.** After training a transcoder or SAE, we generate explanations for all the latents using contexts where that latent is active. An LLM is tasked to find patterns in these activations and output a simple, single-sentence explanation for that latent. These explanations are used by another LLM instance to predict whether the latent should be active on a given token. After post-processing those predictions, we produce a reconstructed vector by pushing the simulated activations through the decoder.

The idea of rewriting a neural net in a more interpretable form is not new. The "microscope AI" framework [Hubinger, 2019] aims to analyze a neural network's learned representations to gain actionable insights for humans, rather than using the network directly. These insights would likely take the form of natural language explanations of the network's features and circuits. Microscope AI aims to reduce risks associated with model deployment while still benefiting from the model's knowledge. Imitative generalization is a proposal to extend this idea by jointly optimizing the network and its human-interpretable annotations to maximize their prior likelihood [Barnes, 2021]. Sparse autoencoders and similar techniques can then be used as "explainer" models, explaining the behavior of the network by sparsely decomposing its activations.

In this work, we pursue the following idea: if the latents of a transcoder are interpretable enough, we can *simulate* its activations using natural language explanations. Specifically, we replace the encoder of the transcoder with an LLM prompted to predict the activation of each latent given its explanation and the textual context. We then patch this modified transcoder back into the model, hopefully yielding behavior nearly identical to the unpatched model. In the limit, we could use this to "rewrite" every feedforward layer in the model in terms of interpretable features and operations on those features.

Unfortunately there are several roadblocks for this approach. Firstly, despite a majority latents seeming to have interpretable activations, a large fraction is either uninterpretable or not well-captured by our current pipeline. Secondly, even the ones that seem interpretable are hard to explain in their totality, as it is hard to generate simple explanations that are both contextually sensitive and specific. Finally, the predictions of the simulator LLM are not calibrated: it over-estimates the frequency with which latents fire, and how strong their activations should be. Because we believe that there are many potential ways to improve the quality and specificity of explanations, we see this test as a potential benchmark of transcoder and SAE interpretability.

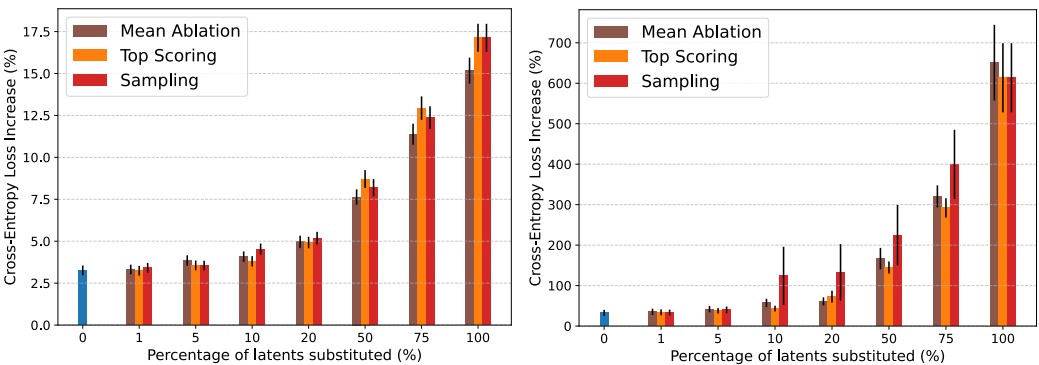

Figure 2: **Cross entropy loss increase for different fractions of transcoder and SAE substitution.**
We compute the CE loss over 10K prompts, for the transcoder (left) and over 1K prompts SAE
(right) respectively, by substituting parts of the encoder with natural language explanations. Bars
in orange show the loss increase when selecting the latents with the highest interpretability scores
for replacement. Bars in red show the loss increase when randomly selecting a subset of latents to
replace. Bars in brown show the loss increase caused by replacing a random subset of the latents
with their mean activations– a simple baseline that we should ideally be able to overcome. The
blue bar represents the increase in loss from using the sparse coders as is. Bar heights represent the
mean value of the increase with respect to the base loss, while error bars represent the standard error.
The interpretability score used for selecting latents is detection scoring, [Paulo et al., 2024, page 5],
computed over 100 positive and 100 negative samples. Over this set of prompts, Pythia had a cross
entropy loss of $3.19 \pm 0.09$ nats per token. Prompts where the loss was lower than $0.1$ nats were
excluded.

## 2 Methods

### 2.1 Sparse coder training

We train different types of sparse coders to evaluate their potential for partial rewriting. We begin by
training a skip-transcoder on the MLP of the sixth layer of Pythia 160M [Biderman et al., 2023] as
well as a skip-transcoder on the MLP of the fifteenth layer of SmolLM2 135M [Allal et al., 2025]. We
also train a sparse autoencoder on the output of the 6th layer of Pythia 160M. The skip-transcoders
have a linear "skip connection," which Paulo et al. [2025] found improves the ability to approximate
the original MLP at no cost to interpretability scores. That is, the transcoder takes the functional form

$$\hat{y} = \mathbf{W}_2 \text{TopK}(\mathbf{W}_1 x + \mathbf{b}_1) + \mathbf{W}_{\text{skip}} x + \mathbf{b}_2 \tag{1}$$

where $x$ is the input of the MLP and $\hat{y}$ is the reconstructed output of the MLP. Both $\mathbf{W}_2$ and $\mathbf{W}_{\text{skip}}$
are zero-initialized, and $\mathbf{b}_2$ is initialized to the empirical mean of the MLP outputs, so that the
transcoder is a constant function at the beginning of training. All models are trained to minimize the
mean squared error $||y - \hat{y}||_2^2$ between the model's output and the target module output.

The sparse coders trained on Pythia have 32768 latents, while the ones trained on SmolLM2 have
18432 latents. Sparsity is continuously enforced on the sparse coder latents using the TopK activation
function proposed by Gao et al. [2024] with $k = 32$ for Pythia and $k = 128$ for SmolLM2.[1] The
sparse coders trained on Pythia are trained over the first 8B tokens of the Pile [Gao et al., 2020], using
the Adam optimizer [Kingma, 2014], a sequence length of 2049, and a batch size of 64 sequences.[2]
The skip-transcoder trained on SmolLM2 was trained on 1B tokens of the FineWeb-Edu dataset
[Lozhkov et al., 2024], part of the training corpus of the model, using the schedule-free Signum
optimizer [Bernstein et al., 2018, Defazio et al., 2024], a sequence length of 2049, and a batch size of
64 sequences. For this model, schedule-free Signum was used because it let to a significantly better
reconstruction loss and a lower number of dead latents.

---

[1]The higher $k$ for SmolLM2 was an attempt to achieve higher rewriting performance by increasing the
number of active latents, after our initial experiments with Pythia at $k = 32$ yielded disappointing results.

[2]The optimizer appeared to converge after 1B tokens, so we early stopped the training run.

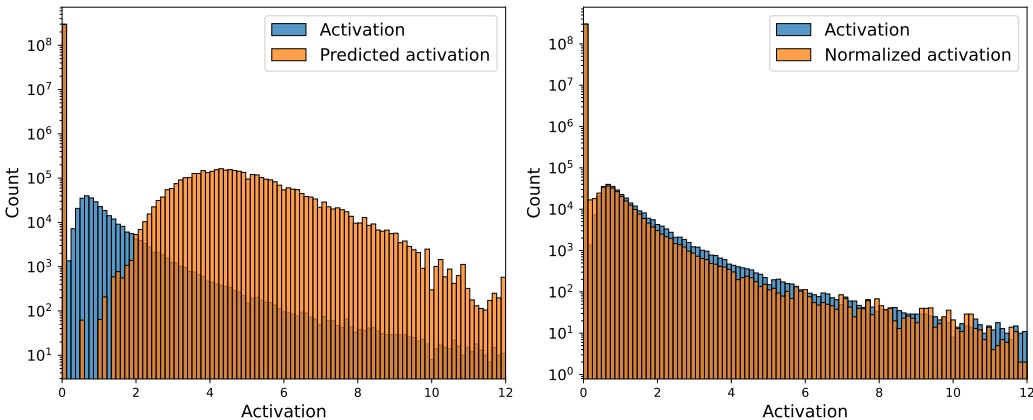

Figure 3: **Distribution of predicted activations for all latents.** On the left we compare the distribution of predicted activations before normalization, and on the right we show what the distribution looks like after quantile normalization. Before normalization, the predictor model systematically over-predicts high activation values by multiple orders of magnitude. Quantile normalization primarily has the effect of enforcing a prior in favor of features not being active.

## 2.2 Interpreting latents

We use the automated interpretability pipeline released by Paulo et al. [2024] to generate explanations and scores for transcoder and SAE latents. For all sparse coders we collected activations over 10M tokens of the same dataset they were trained on. For each latent, representative samples of its activations are sampled and shown to an LLM, in our case Llama 3.1 70b Instruct [Dubey et al., 2024], which is told to give a succinct explanation that summarizes the activations (Figure 1, first row). We only explain latents that are active on more than 200 instances over the 10M samples taken. The LLM is shown 40 examples, 4 examples from each of the 10 activation deciles for that latent.

After explanations are generated for each latent, they are scored. We also use the automated interpretability pipeline from Paulo et al. [2024] for this process. We use both fuzzing and detection to score the latents. To compute the detection score, Llama 3.1 70b Instruct is given the explanation of the latent and a set of examples. The LLM then has to decide which examples activate the latent and which don't using the explanation that was given. We repeat this process 20 times, showing five examples each time, totaling 50 non-activating examples and 50 activating examples. We use stratified random sampling from the 10 activating deciles of that latent. At the end the detection score of the latent is given by the $F_1$ score of the scorer, $F_1 = \frac{2}{p^{-1}+r^{-1}}$, where $p$ is the precision and $r$ is the recall. The fuzzing score is computed with a similar protocol, but instead the LLM is shown a set examples where one token is highlighted in each, and the model has to label each highlighted token as activating or non-activating. Again the final metric used is the $F_1$ score.

## 2.3 Simulation

The next step in the pipeline is to use an LLM to simulate the activation of a latent in a context given the latent's explanation (Figure 1, middle panel). For this task, we select up to 10K sequences, each 32 tokens long, and highlight the final one. The LLM is tasked to output a single number from 0 to 9 to quantify the activation of the latent on that token for that specific sequence. Because our scorer model has a token for each independent digit, we can estimate the expected value by summing the probability of each digit times the value of the digit. We then map this number to the real activations by dividing the expected value by nine (yielding a number between 0 and 1), then multiplying this value by the maximum observed activation for the latent. The prompt used for simulation can be found in Appendix A.

### 2.3.1 Quantile normalization

We found in early experiments that Llama produces highly uncalibrated predictions of feature activations: the marginal distribution of the predicted activations differs markedly from the marginal distribution of the true activations (Figure 3, left panel). Patching these uncalibrated activations into the model yields very poor results. To alleviate this problem, we use quantile normalization, which monotonically transforms the model's predictions in such a way that their marginal distribution matches that of the true activations. This transformation is an optimal transport map under a variety of cost functions [Santambrogio, 2015].

We compute the quantile normalizer separately for each individual feature, using the empirical CDFs of the simulator's predicted activations and the true activations of a transcoder computed from a random sample of 10% of the data. Once the quantile normalizer has been computed, this transformation is then applied to all simulator predictions. This transformation successfully reduces the number of predicted active latents to numbers that are coherent with the true distribution.

## 3 Evaluation

We evaluate the rewriting of the model by measuring the resulting cross-entropy loss in next token prediction on chunks of text sampled from the Pile. These chunks of texts have lengths that are uniformly sampled from 32 to 256 tokens. This range was selected such that, at its minimum, there is a significant number of tokens that can be shown to the simulator model, and at its maximum the sequences are short enough that the presence of very long range features is unlikely.

Simply replacing a single MLP with a transcoder increases the model's cross-entropy loss by about 10%. This loss increase is equivalent to that of using an early Pythia checkpoint, one that was trained on only 25% of the data,[3] rather than the final Pythia checkpoint. Rewriting any part of the transcoder in natural language will necessarily degrade the model's performance even further. Consequently, we focus on rewriting a single MLP block, since rewriting all MLP blocks simultaneously would cause the model to become completely unusable. The same applies to the residual stream SAE: the cross-entropy loss of the model when adding a single SAE to the residual stream increases by 33%. This is equivalent to using a Pythia checkpoint trained on less than 10% of the data.

After collecting the predicted activations for all features of a given token in a given sentence, and after applying quantile normalization to those predictions, we use the TopK activation function to ensure it has the expected level of sparsity. This activation vector is fed into the decoder, yielding a reconstructed output vector. In the case of the transcoders, the original output of the MLP is discarded and in its place we inject the reconstructed vector (Figure 1), while in the case of the SAE the output of the layer is discarded and in its place the reconstructed vector is patched in.

We evaluate over 10K different contexts for the Pythia transcoder, 2K for the SmolLM2 transcoder, and 1K for the Pythia SAE, measuring the cross-entropy loss for next-token prediction. This process is very expensive, as each context requires individually prompting the simulator for 32 thousand latents. We therefore had to make a total of 327 million queries to the simulator model. Our simulator models were hosted locally using VLLM [Kwon et al., 2023], and the experiments took 1 week on 4 nodes with 8xA40 GPUs.

**Partial rewriting.** We also experiment with mixing predicted and ground truth latent activations in varying proportions, allowing us to examine the effect of rewriting only part of the encoder. We do this in two different ways:

1. Select the top $k$ most interpretable features according to our evaluation pipeline [Paulo et al., 2024]. This is labeled "Top scoring" in Figure 2.

2. Sample $k$ features uniformly at random from the transcoder. This is labeled "Sampling" in Figure 2.

The "Mean Ablation" baseline condition is like "Sampling" except that we replace the selected latents with their mean activations.

---

[3]The cross-entropy loss of Pythia 160M checkpoints on this set of prompts is not monotonic with training time, so a more precise estimate is not possible.

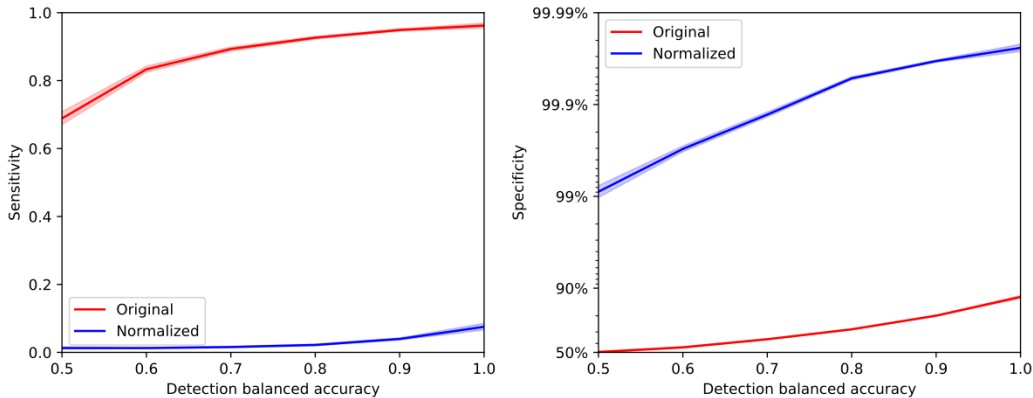

Figure 4: **Detection score predicts sensitivity and specificity.** Binning explanations by their scores makes it evident that high-scoring explanations are more specific and sensitive.

## 4 Results

The left panel of Figure 2 illustrates how cross-entropy loss increases on Pythia as we replace more and more transcoder latents with their simulated counterparts. Unfortunately, we see that the "Top Scoring" and "Sampling" conditions both result in very similar loss increases to the baseline "Mean Ablation" condition, for most fractions of substitution. The same behavior can be seen when using the residual stream SAE (Figure 2, right panel), and when using the transcoder trained on SmolLM 2 (Figure A2). For some fractions and models we observe that randomly selecting which latents to substitute, instead of always substituting the most interpretable latents, leads to a smaller increase in the cross entropy loss, showing that the interpretability metrics are somewhat capturing how easy it is to describe the behaviour of the latents. While our substitution method works better than mean ablation for some fractions, it is not consistently better and in fact when we do full substitution we find that there is almost no difference between the two.

This indicates that our explanation pipeline is incapable of rewriting even a small fraction of a transformer without causing a substantial increase in the next-token prediction loss– an increase that is comparable to simply replacing neurons in the transformer with their mean values. This is a negative result which suggests more work needs to be done to improve explanation quality.

### 4.1 Explanations are not detailed enough

One of the reasons that we need to calibrate the activation predictions is because we find that current explanations do not allow the model to be specific enough. To understand this, let's consider the case that the classifier only achieves a specificity of 99%. The transcoder used in this work has 32768 latents, and if the LLM predictor can only achieve a specificity of 99% this means that, that on average, it predicts that there are 320 active latents, 10 times more than the actual number ($k = 32$). Even with a sensitivity of 100%, where the model correctly identifies all of the latents that should be active, it is unlikely that the top 32 predictions all would be the correct ones. We observe than on average the current automatic latent explanation setup has a specificity of around 80%, meaning that on average the model identifies close to 6 thousand latents as being active.

By performing quantile normalization, a large chunk of incorrectly predicted activations, activations that should be zero but were given a non-zero value by the predictor, are set back to zero. This significantly increases the specificity, enough that some of the original model's performance is maintained, but at the same time this significantly decreases the sensitivity, as some of the correctly classified active latents are also set to zero.

We find that detection scores [Paulo et al., 2024, page 5] are predictive of the specificity and sensitivity of an explanation, with higher scoring latents corresponding to explanations that have higher specificity and sensitivity (Figure 4). This is expected, as detection scoring corresponds to detecting whether a given latent is active on a given context, which is similar to our simulation task

in this work. For the same reason, latents with higher fuzzing scores also have higher sensitivity and specificity (Figure A1).

# 5  Limitations

We divide the limitations of this work into three categories. One type of limitation is related with the quality of sparse coders we investigated. We only explored TopK sparse coders, trained on one to ten billion tokens, with an expansion factor of 32x. However, we don't see any evidence in the literature that other activation functions would make the sparse coders more interpretable, so it is unlikely that other architectures would significantly improve our results. Larger sparse coders would lead to lower reconstruction losses but would require even higher specificity.

The quality of explanations is a limitation of our work, as it is possible that better explanation generation techniques could yield better rewriting results. While the quality of explanations could likely be improved with careful prompt tuning or even finetuning the explainer model, we think the results would not substantially change due to the huge gap between the specificity of current explanations and the specificity that would be required for satisfactory model rewriting.

Rewriting is also very computationally expensive. Larger models would probably be able to simulate the activations more accurately, but this would make the process even more expensive.

# 6  Conclusion

In this work, we proposed a new methodology for rigorously evaluating the faithfulness of natural language explanations of sparse coder latents in transformers, based on partially rewriting the transformer using these explanations. We found that existing explanations are severely wanting. We are still unable to outperform mean ablation, where each latent is substituted by its average value over the dataset.

This is mainly due to the fact that explanations are not specific enough, leading to a high number of false positives. Our results highlight the fact that it is important for an explanation to correctly identify the contexts where a feature is not active, in addition to the feature's activation level in contexts where it is active. Future work on the interpretability of latents should take this into consideration.

To improve upon these results, new techniques are needed to make explanations more specific, for instance using contrast pairs of highly similar features to bring out additional details. This could potentially increase the sensitivity as well, which takes a big hit when using quantile normalization.

## Impact Statement

This paper presents work whose goal is to advance the field of Mechanistic Interpretability. There are many potential societal consequences of our work, none which we feel must be specifically highlighted here.

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

 ## A  Simulation prompt

```
You are an intelligent and meticulous linguistics researcher.

You will be given a  certain explanation of a feature of
text, such as "male pronouns" or "text with negative sentiment"
and examples of text that contains this feature. Some explanations
will be given a score from 0 to 1. The higher the score the better
the explanation is, and you should be more certain
of your response (positive or negative).

These features of text are normally identified by looking for specific
words or patterns in the text. There are many features associated
with a single token, and sometimes the feature is related with
the previous token or context.

Your job is to identify how much the last token,  which is marked
between << and >>, represents the feature.  You will output
a integer between 0 and 9, where 0 corresponds to no relation
to the explanation and 9 to a strong relation.

Most of the tokens should have no relation. The ones that
are related, should more likely be given 1 than 2, 2 than 3,
and so on. Only give a 9 if the  description exactly matches the token.

You must return your response in a valid Python list.
Do not return anything else besides a Python list.
```

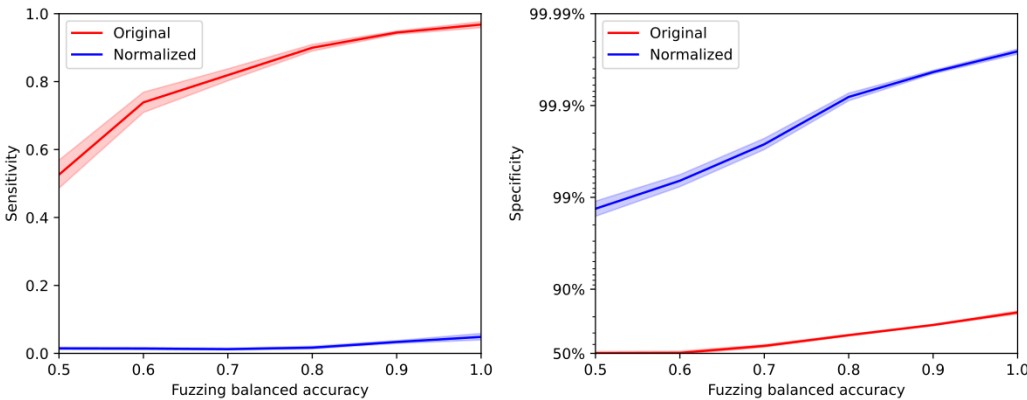

Figure A1: **Fuzzing score predicts sensitivity and specificity** Explanations with higher fuzzing scores lead to better predictions of the simulations

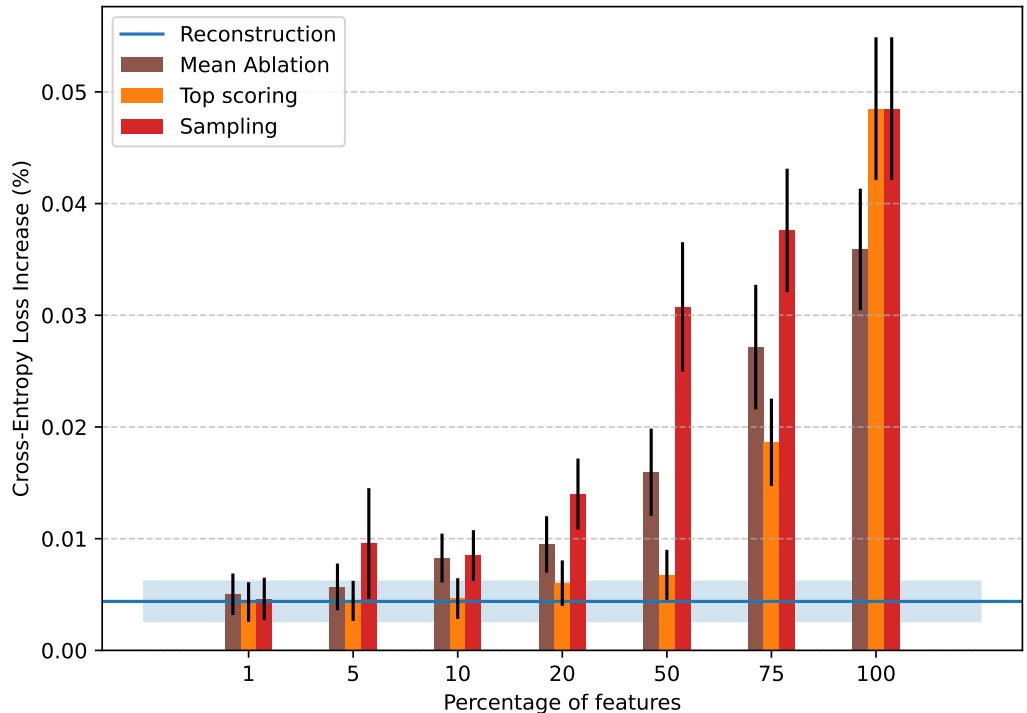

Figure A2: **Cross entropy loss increase for different fractions of transcoder in SmolLM 2.** We compute the CE loss over 2K prompts, for the transcoder, by substituting parts of the encoder with natural language explanations. Bars in orange show the average loss increase when choosing the top scoring latents for replacement. Bars in red show the average loss increase when randomly selecting a subset of latents to replace. Bars in brown show the average loss increase caused by using the mean activations out a part of the transcoder. Bar heights represent the mean value of the increase with respect to the base loss, error bars represent the standard error. The interpretability score used for the selecting latents is detection scoring, [Paulo et al., 2024, page 5], computed over 100 positive and 100 negative samples. Prompts where the loss was lower than 0.1 nats were excluded.

