# OpenReview forum: "Can We Partially Rewrite Transformers in Natural Language?"
_NeurIPS.cc/2025/Conference — Submitted to NeurIPS 2025_

### Official Review · Reviewer_VTXa · 2025-06-20

**Clarity:** 3
**Significance:** 1
**Originality:** 1
**Rating:** 3
**Confidence:** 4

**Summary:**

The authors propose a methodology to test whether recent findings in the field of mechanistic interpretability allow for ‘rewriting'  large language models (LLMs) to arrive at their predictions in a more interpretable manner.

Specifically, recent work has found that training sparse auto-encoders (SAEs) on the internal representations of LLMs can lead to (more) disentangled representations in which individual neurons inside the SAE learn to consistently fire when a semantically meaningful concept is present in the model input.

Building on this, the authors propose to (1) automatically give those neurons a human-understandable short explanation by prompting an LLM to summarize some example activations for those neurons, then (2) use a second LLM to predict whether for a new input and a set of explanations a given neuron is likely to fire , and (3) finally, to then use the thus obtained expected firing patterns to update the model’s internal activations as if those neurons had fired.

In their experiments, the authors find little evidence that this approach allows for rewriting LLMs.

**Questions:**

Summarizing my above concerns, my main questions / recommendations to the authors are as follows.

1. (Weakness 1) Is there a good reason to use an LLM as a judge to predict the SAE activations?
2. (Weakness 1) The submission would benefit from more clearly distilling what the findings are; the current conclusions along the lines of "we find that current sparse coding techniques and automated interpretability pipelines are not up to the task of rewriting even a single layer of a transformer" are too general given the experimental setup.
3. (Weakness 2) If the authors find that the LLM is not able to predict the SAE activations, why would one expect that the only problem of the LLM is merely over-predicting the activations and that monotonous rescaling would yield anything useful?
4. (Weakness 3) How can we trust the SAE training to have been 'successful' in the sense that it yields similar results to those reported in the literature? Without establishing this, any statement about SAEs that goes beyond this specific study is hard to justify given the fact that SAEs are difficult to train properly.
5. (Weakness 1-4) In general, I do not find the experimental setup very convincing and would highly appreciate if the authors could clearly motivate why this specific approach would make sense to explore.
6. (Weakness 5) I would appreciate if the authors could more clearly explain what aspects of their work they believe to be the root cause of the negative results, in particular in relation to other works that have shown promising results when it comes to either activation patching or sparse autoencoders.

**Ethical Concerns:**

["NO or VERY MINOR ethics concerns only"]

**Final Justification:**

The authors were unfortunately not able to address my concerns during the discussion.

In particular, the over-generality of the submission's claim in combination with the ad-hoc nature and the limited motivation for the proposed approach do, in my opinion, not meet the bar for a NeurIPS submission.

That said, I understand that other reviewers have a more favorable view of the submission and I am curious to hear their arguments in favor of accepting the submission for publication.

**Limitations:**

yes

**Quality:**

1

**Strengths And Weaknesses:**

**Strengths**

- The submission is clearly written and thus easy to read.
- The submission highlights some important related works in the field of mechanistic interpretability, which help motivate the study.
- The authors evaluate their method along various design choices (autoencoder vs transcoder, SmolLM vs Pythia, various ‘patching/rewriting’ techniques).
- The authors clearly highlight shortcomings of their approach and openly discuss their negative results.

**Weaknesses**

Most importantly, I find the design proposed by the authors for rewriting LLMs to lack sufficient motivation and principled foundations. Unfortunately, this makes it very hard to draw generalizing conclusions from this work. While the authors conclude based on their study that “[they] find that current sparse coding techniques and automated interpretability pipelines are not up to the task of rewriting even a single layer of a transformer” (see abstract), I find it hard to draw a conclusion that goes beyond “This specific design for patching the activations in transformers significantly degrades the output of the transformer as measured in cross-entropy”. More specifically, I have the following concerns.

1. **Full and ill-motivated reliance on LLMs, unclear failure points.** The authors follow a recent trend of heavily relying on a fully automated LLM pipeline to handle most of the data processing, from labelling the neurons to judging whether the neurons should be active. While this hands-off approach naturally scales more easily, it arguably just compounds the problem of interpretability by explaining one LLM with the predictions of a potentially unreliable second LLM. Moreover, it remains unclear to me why one would even want to use an LLM-as-a-judge in the first place to predict the activation of the SAEs. If the underlying assumption is that the SAE on its own already captures interpretable concepts (which seems to be the authors’ assumption), it would make more sense to just use the SAE as is. On the other hand, if the SAE is not expected to capture interpretable concepts sufficiently well, I do not see how adding an LLM on top would add any benefit whatsoever, in particular because the SAE’s learnt reconstructions are used to patch the original model; as far as I see it, the second LLM can only introduce additional failure points. **In summary,** while I find it laudable that the authors openly discuss their negative results, the submission does not allow for gaining novel insights into why this pipeline does not work.
2. **Rescaling distributions.** The authors find the LLM-as-a-judge to be a bad judge (underlining my previous point), yielding highly different activation distributions from the SAE itself. While it already seems questionable to expect that the LLM would, without any training whatsoever, be able to predict the SAE activations, simply forcing the distributions of actual and LLM-predicted firing patterns to ‘look’ the same (Fig. 3) lacks a proper motivation; in fact, the difference in distribution should rather be taken as a sign of caution that the LLM-as-a-judge is not well-suited for the task.
3. **Evaluating the SAE.** Training SAEs properly can be a difficult task (cf. “Scaling and evaluating sparse autoencoders”, Gao et al 2024) and, given the authors’ current evaluations, it remains unclear to me how well the SAE training even worked in the first place. Specifically, the submission currently lacks any comparisons in terms of the SAE training that would allow the reader to conclude that the SAE is ‘state of the art’ and that it captures any interesting concepts in the first place. Without establishing that the SAE generally performs similarly to reported results in the literature, it is difficult to identify the failure points of the proposed approach.
4. **Utility and evaluation of the proposed framework.** As it stands, the utility of the proposed framework as well as how to evaluate it beyond a decrease in cross-entropy loss remains unclear to me. This is particularly worrisome given how expensive the proposed approach is (see lines 145-150).
5. **Limited discussion of related work.** While the authors select some impactful related work examples to motivate their study, they do not adequately contextualize their work by discussing other relevant related works. E.g., recent works (e.g. [Patchscopes](https://arxiv.org/abs/2401.06102) by Ghandeharioun et al. (2024)) have shown that, when done carefully, activation patching can indeed lead to interesting insights. In contrast to this, the authors make a very broad claim that current techniques cannot even explain a single transformer layer. The submission would significantly benefit from discussing similarities and differences to such works to understand why they believe that current techniques seem to generally not be 'up to the task'.

**Minor**.
In addition to the above points, there are many smaller design choices that are not motivated well (e.g. evaluation of 6th layer of Pythia, 15th layer of SmolLM2; sampling explanations across percentiles, number of examples shown to the LLM). This gives an overall impression of the proposed pipeline to be based on a sequence of ad-hoc decisions.

---

> ### Author Rebuttal · Authors · 2025-07-30
>
> > (Weakness 1) Is there a good reason to use an LLM as a judge to predict the SAE activations?
>
> Yes. Explanations are meant to enable a human or an LLM to better predict the activations of an SAE. Other works have shown that LLMs are about as good as humans at generating explanations, and at using those explanations to make predictions about whether latents are active or not in a given context. It would be infeasible to use human labor for this task, and using an LLM ensures consistency of the predictions across features and examples.
>
> > (Weakness 1) The submission would benefit from more clearly distilling what the findings are; the current conclusions along the lines of "we find that current sparse coding techniques and automated interpretability pipelines are not up to the task of rewriting even a single layer of a transformer" are too general given the experimental setup.
>
> As we stated in response to reviewer TeiC, there is a definite upper limit to explanation quality, as long as we restrict ourselves to reasonably short explanations. In _Interpretability as Compression_, Ayonrinde et al. (2024) emphasized the importance of restricting or penalizing the length of an explanation when evaluating its quality, since a sufficiently long and detailed explanation could perfectly capture the activation pattern of a feature in a trivial way. Hence, while our pipeline is imperfect, we do not believe there is sufficient room for improvement to completely overturn our negative results.
>
> > (Weakness 2) If the authors find that the LLM is not able to predict the SAE activations, why would one expect that the only problem of the LLM is merely over-predicting the activations and that monotonous rescaling would yield anything useful?
>
> We directly show in Figure 3 (left panel) that the LLM dramatically over-predicts the activation values before quantile normalization is applied. We saw this in early experiments and deduced that quantile normalization would fix this problem— which it indeed does (Figure 3, right panel).
>
> > (Weakness 3) How can we trust the SAE training to have been 'successful' in the sense that it yields similar results to those reported in the literature? Without establishing this, any statement about SAEs that goes beyond this specific study is hard to justify given the fact that SAEs are difficult to train properly.
>
> We use the standard, widely used open source library called sparsify (https://github.com/EleutherAI/sparsify) to perform our transcoder training. Its default hyperparameters are known to produce reasonably high quality sparse coders. We can update the manuscript with numbers related to the final reconstruction loss, but the increase of CE loss shown in Figure 2, blue bar, should be sufficient evidence that the SAE was trained ‘successfully’, as these numbers are similar to the ones reported in the literature.
>
> > (Weakness 1-4) In general, I do not find the experimental setup very convincing and would highly appreciate if the authors could clearly motivate why this specific approach would make sense to explore.
>
> Automated interpretability using an LLM as a judge is a well-established technique for evaluating the interpretability of SAE features; see for instance Scaling Monosemanticity: Extracting Interpretable Features from Claude 3 Sonnet by Templeton et al. (2024) and Automatically Interpreting Millions of Features in Large Language Models by Paulo et al. (2024). Our experimental setup is also grounded in the stated goals of the mechanistic interpretability field, which has always aimed at reverse engineering neural networks. For example, the 2024 ICML workshop, accessible at https://icml2024mi.pages.dev/, states: “One emerging approach for understanding the internals of neural networks is mechanistic interpretability: reverse engineering the algorithms implemented by neural networks into human-understandable mechanisms.”
>
> > (Weakness 5) I would appreciate if the authors could more clearly explain what aspects of their work they believe to be the root cause of the negative results, in particular in relation to other works that have shown promising results when it comes to either activation patching or sparse autoencoders.
>
> No prior work has performed experiments similar to ours, so we do not believe that our negative results are in tension with prior work that may seem more positive about the prospects of sparse coders. While previous work on transcoders and SAEs show that one is often able to extract superficially interpretable features from LLM activations, features are rarely completely interpretable, and in fact a significant fraction of them can’t be easily understood. Our results stress tested the assumption that latents are easily interpreted by looking at their activation patterns. We believe that our negative results stem from deep facts about how large language models work. There is no guarantee that the collection of all of the features of transcoders and SAEs, filtered through their natural language explanations, can capture all the information in the original activations. Explanations are always incomplete, and sparse coders almost always have non-negligible reconstruction errors. It is likely true, as Hubert Dreyfus argued starting in the 1960s, that intelligence— whether natural or artificial— cannot be decomposed into a list of independent atomic features.

---

> > ### Comment · Reviewer_VTXa · 2025-08-03
> >
> > Thank you for taking the time to respond to my concerns. Unfortunately, I am still not convinced about the conclusions drawn and the generality of the authors' claims and still do not understand the motivation for the proposed approach. More detailed comments follow.
> >
> > > Yes. Explanations are meant to enable a human or an LLM to better predict the activations of an SAE. Other works have shown that LLMs are about as good as humans at generating explanations, and at using those explanations to make predictions about whether latents are active or not in a given context. It would be infeasible to use human labor for this task, and using an LLM ensures consistency of the predictions across features and examples.
> >
> > This does not answer my question regarding why one would want to predict the activations of the SAE instead of using the activations themselves. There is no benefit to the overall explanation pipeline as far as I can see and it just adds more randomness and less interpretability to the pipeline (given that a black box model is used to predict activations of latents). I still do not find this aspect convincing.
> >
> > > As we stated in response to reviewer TeiC, there is a definite upper limit to explanation quality, as long as we restrict ourselves to reasonably short explanations. In Interpretability as Compression, Ayonrinde et al. (2024) emphasized the importance of restricting or penalizing the length of an explanation when evaluating its quality, since a sufficiently long and detailed explanation could perfectly capture the activation pattern of a feature in a trivial way. Hence, while our pipeline is imperfect, we do not believe there is sufficient room for improvement to completely overturn our negative results.
> >
> > Unfortunately, the submission provides little evidence to back up the authors' belief that there is insufficient room for overturning their negative results. I agree with reviewer 6nTt on this point and am still not convinced that the presented ad-hoc pipeline can be generalised to the extent that the authors do. What this paper shows is that the specific pipeline that the authors chose to use does not yield satisfactory results. Given that the motivation of the pipeline is unclear to me, the relevance and insights gained from this submission do not meet the bar for publication at NeurIPS.
> >
> > > We directly show in Figure 3 (left panel) that the LLM dramatically over-predicts the activation values before quantile normalization is applied. We saw this in early experiments and deduced that quantile normalisation would fix this problem— which it indeed does (Figure 3, right panel).
> >
> > The normalisation is by design scaling the overall distributions such that they overlap. This, however, does not mean that individual predictions are correct and it remains unclear how well the LLM is able to solve this task.
> >
> >
> > > Automated interpretability using an LLM as a judge is a well-established technique for evaluating the interpretability of SAE features; see for instance Scaling Monosemanticity: Extracting Interpretable Features from Claude 3 Sonnet by Templeton et al. (2024) and Automatically Interpreting Millions of Features in Large Language Models by Paulo et al. (2024). Our experimental setup is also grounded in the stated goals of the mechanistic interpretability field, which has always aimed at reverse engineering neural networks. For example, the 2024 ICML workshop, accessible at https://icml2024mi.pages.dev/, states: “One emerging approach for understanding the internals of neural networks is mechanistic interpretability: reverse engineering the algorithms implemented by neural networks into human-understandable mechanisms.”
> >
> > Yes, LLM-as-a-judge techniques have and can indeed be used to automate data labelling, which of course includes explanations. However, the specific usage of this technique by the authors is not well motivated; again, why use an LLM to predict the activity of sparse activations which one has the SAE for in the first place.

---

> > > ### Author Response · Authors · 2025-08-04
> > >
> > > > This does not answer my question regarding why one would want to predict the activations of the SAE instead of using the activations themselves.
> > >
> > > The purpose of predicting SAE activations is to evaluate the quality of our explanations. Without a prediction step, we would have no way of knowing if the explanations are genuinely capturing the activation pattern of the SAE feature. Many SAE features are simply uninterpretable, and in these cases the explainer model either produces a very general explanation or hallucinates some pattern that is not really there. We detect these low-quality explanations by forcing an LLM to predict the activation of the SAE feature in unseen contexts, with reference to the explanation. If the explanation is poor, the LLM will perform poorly at this prediction task. By contrast, a good explanation will enable the LLM to predict the SAE activation with high accuracy.
> > >
> > > This approach is not new. It has been used in several prior works, originally in the context of neuron activations, and later in the context of SAE activations. The following is an incomplete list:
> > > - Bills et al. (2023) pioneered the simulation approach (using raw neurons rather than SAE features) in their paper _Language models can explain neurons in language models_.
> > > - In _Towards Monosemanticity_, Bricken et al. (2023) suggested that this same approach could be used for SAEs, but did not run the experiments.
> > > - Auto-interp was then used on SAE activations by Templeton et al. (2024) in their paper _Scaling Monosemanticity: Extracting Interpretable Features from Claude 3 Sonnet_.
> > > - In _Automatically Interpreting Millions of Features in Large Language Models_, Paulo et al. (2024) introduce an open source pipeline for auto-interp. Their pipeline includes multiple different techniques for evaluating the quality of SAE feature explanations, including LLM-based simulation.
> > > - In _Scaling Automatic Neuron Description_, Choi et al. (2024) use language models to predict the activations of neurons, with reference to natural language explanations of those neurons.
> > >
> > > All of these papers use LLMs to predict activations, and use the accuracy of these predictions as a measure of how good an explanation is, and thereby indirectly measure how interpretable the neurons or SAE features themselves are. We are merely extending this existing body of work.
> > >
> > > > There is no benefit to the overall explanation pipeline as far as I can see and it just adds more randomness and less interpretability to the pipeline
> > >
> > > Prediction is an essential part of the pipeline— it is a quality control and evaluation step. Without it, we would have no way of knowing how well we are doing, how good the explanations are, and how well the SAEs are being trained. Without prediction, our explainer model could simply output garbage explanations with no relation to the actual SAE features it is trying to explain, and we would be none the wiser.
> > >
> > > > Unfortunately, the submission provides little evidence to back up the authors' belief that there is insufficient room for overturning their negative results.
> > >
> > > Our most general claim is: ”We find that current sparse coding techniques and automated interpretability pipelines are not up to the task of rewriting even a single layer of a transformer”, which we believe is adequately hedged. We don’t claim that this task is impossible, but only that current interpretability techniques are not good enough. There are a handful of sparse coder architectures in use today, but all of them are found to be roughly equally interpretable, so using a different sparse coder would unlikely give different results. The interpretability pipeline that we used is, as far as we are aware, the open source state of the art.
> > >
> > > > The normalisation is by design scaling the overall distributions such that they overlap. This, however, does not mean that individual predictions are correct and it remains unclear how well the LLM is able to solve this task.
> > >
> > > This is exactly right. We show in the paper that the LLMs do poorly at predicting the activations given an explanation, even after quantile normalisation. Our claim is precisely that LLMs are unable to solve this task adequately, even after applying a fairly aggressive statistical normalisation technique to their predictions. Far from being a flaw in our paper, this is exactly what we intended to show.
> > >
> > > > the specific usage of this technique by the authors is not well motivated; again, why use an LLM to predict the activity of sparse activations which one has the SAE for in the first place.
> > >
> > > The purpose of predicting the activity of sparse activations is to evaluate the quality of our explanations. We are not intending to use the predicted activations in any kind of inference pipeline. Our paper is entirely about evaluation and quality control. SAEs on their own, without any explanations, do not serve any purpose. Hence, it is essential to both generate explanations and evaluate their quality.

---

### Official Review · Reviewer_TeiC · 2025-06-30

**Clarity:** 3
**Significance:** 3
**Originality:** 3
**Rating:** 4
**Confidence:** 4

**Summary:**

The paper evaluates the quality of LLM-generated explanations of SAE latents. Specifically, they propose to prompt an LLM to predict the activation values of the SAE latents on new text given the generated explanations, patch those simulated activations back into the network, and observe the effect on cross-entropy loss. They find that substituting even a fraction of activations degrades the model as much as mean ablation, regardless of whether the most interpretable latent or random ones are chosen. They also observer that Llama predictions generally overestimate activation values and because explanations lack specificity, it often marks thousands of latent as active instead of the expected 32 latents. Overall, the authors find that explanation pipelines are not up to the task of rewriting even a single layer.

**Questions:**

- Have you experimented with few-shot prompts for the simulation? I would expect the calibration to be better if the model would have been presented a couple examples.
- You mention that you only explain latents that are active on more than 200 instances (L89), but report substitution of latents up to 100% in Figure 2. Does that mean you found more than 200 instances for all latents, or does the 100% in Figure 2 refer to all latents for which you were able to generate explanations? If the latter, how many latents were not active on at least 200 instances?

**Ethical Concerns:**

["NO or VERY MINOR ethics concerns only"]

**Final Justification:**

Thank you for your response. I have read the other reviews and author rebuttals.

While SAEs have been established as one of the default techniques for finding interpretable features in neural network representations and LLMs have been used to generate natural language explanations for SAE latents, these explanations have well known limitations. However, most work on evaluating the quality of generated explanations relies on human judgement or focuses on a small selection of SAE latents. I consider the experiments in this paper an interesting approach to scale the evaluation of those natural language explanations. Thus, despite its high cost and limitations (including confounders), I believe it could be a useful result and methodology for the field of interpretability.

**Limitations:**

- The evaluation is limited to two small language models; Pythia-160 M and SmolLM-2 (125 M). Smaller models tend to learn lower-quality representations than their larger counterparts, so performance might improve with a larger model.

**Quality:**

3

**Strengths And Weaknesses:**

**Strengths:**
- The paper presents an interesting and original perspective, treating "partial rewriting" as a function-preservation benchmark, not just a subjective interpretability score
- The authors openly report failures and analyse them

**Weaknesses:**
- The evaluation is very expensive. The authors mention that their experiments required 327 million queries or one week on 8x A40s. This will likely hinder adoption of the benchmark.
- The evaluation mixes confounding factors: the quality of the SAE/transcoder and the quality of the explanations, which in turn depend on the quality of the LLM.

---

> ### Author Rebuttal · Authors · 2025-07-30
>
> > Have you experimented with few-shot prompts for the simulation? I would expect the calibration to be better if the model would have been presented a couple examples.
>
> Our prompt currently shows 3 example sentences and their real activation values, but it is possible that having a larger number of few shot examples, which allowed the model to estimate the average firing frequency would help, although that would mean that on 100 examples shown, none should be active.
> We also noticed that explicitly telling the model that latents fire very infrequently helped make the predictions better calibrated.
>
> > You mention that you only explain latents that are active on more than 200 instances (L89), but report substitution of latents up to 100% in Figure 2. Does that mean you found more than 200 instances for all latents, or does the 100% in Figure 2 refer to all latents for which you were able to generate explanations? If the latter, how many latents were not active on at least 200 instances?
>
> When reporting 100% in Figure 2 we mean 100% of the ones that fire more than 200 instances. Around 0.5% of features don’t fire more than 200 times.
>
> > The evaluation is limited to two small language models; Pythia-160 M and SmolLM-2 (125 M). Smaller models tend to learn lower-quality representations than their larger counterparts, so performance might improve with a larger model.
>
> We agree this is an important limitation of our study. However, a significantly larger language model would have required a concomitantly larger computational budget. Our pipeline is, unfortunately, already quite expensive as-is.

---

### Official Review · Reviewer_q1xE · 2025-07-01

**Clarity:** 3
**Significance:** 2
**Originality:** 4
**Rating:** 4
**Confidence:** 4

**Summary:**

This work performs a creative evaluation of the quality of LLM-generated explanations of SAE and skip-transcoder latents. In the standard auto-interp pipeline of Bills et al. and others, one uses an "explainer" language model to generate natural language explanations of when a latent fires, and then a "simulator" LLM is given this explanation and tasked with predicting when each latent will fire. The accuracy of the simulator can then be measured, giving us an automated evaluation of the interpretability of any given latent. In this work, the authors attempt to replace the SAE encoder with calls to the simulator LLM, given natural language explanations of every latent in the SAE. They then measure how this "model" can be patched into the LLM's activations. While one can substitute the original SAE for the LLM activations (at a single layer) and largely recover performance, using the LLM simulator model for the SAE encoder substantially degrades performance -- as much as replacing each latent with its mean activation. This is a negative result that suggests that existing pipelines for automatically interpreting model latents are not good enough to capture the model's computation.

**Questions:**

1. Do you think that your overall negative result is more likely to reflect the limitations of auto-interp or the limitations of SAEs themselves? I wonder whether, if we had stronger LLMs, which could, as agents, run experiments to test different hypotheses about latent explanations, or if we gave the simulator and explainer model other information (for instance about other internal latents in earlier network layers), whether something like this could eventually work?

**Ethical Concerns:**

["NO or VERY MINOR ethics concerns only"]

**Final Justification:**

The reviewers responded to my questions, which were mostly philosophical. I've kept my score the same.

**Limitations:**

yes

**Quality:**

3

**Strengths And Weaknesses:**

### Strengths
* This is a very creative way of evaluating the quality of the standard explainer-simulator auto-interp pipeline!
* The title of the paper, and its core idea, remind me of the "What does it mean to understand a neural network?" paper from Lillicrap & Kording, which speculated: "One day, we may develop ways of compactly describing how neural networks work after training. There may be an intermediate language in which we could meaningfully describe how these systems work." Typically, interp people attempt to use a transcoder or a symbolic program or some other object as the intermediate language, but it is interesting to think about how natural language could be used. I appreciate the paper's ambition and framing.
* The basic results are clear. Figure 2 contains the primary result of the paper.

### Weaknesses
* The presentation in the paper is a little informal, and its readability could be somewhat improved. For instance, I'd suggest including titles on the left and right panels in Figures 2 and 3, or using bold when you refer to the **left** or **right** subplots.
* The main result of the paper is not particularly deep. I appreciated the quantile normalization analysis and also Section 4.1 on the specificity of the explanations. But I do wish that there was a little bit more of a takeaway. Do you have any guesses about *why* the current auto-interp pipeline fails? Ideas for how to improve it? It would be ideal if the paper had a bit more to say. What else can you say about what this result means for auto-interp or for interpretability more broadly?

---

> ### Author Rebuttal · Authors · 2025-07-30
>
> > The presentation in the paper is a little informal, and its readability could be somewhat improved. For instance, I'd suggest including titles on the left and right panels in Figures 2 and 3, or using bold when you refer to the left or right subplots.
>
> Thank you for this suggestion. We will be sure to add titles to those subpanels and use bold to make it clear which subpanel we are referring to.
>
> > The main result of the paper is not particularly deep. I appreciated the quantile normalization analysis and also Section 4.1 on the specificity of the explanations. But I do wish that there was a little bit more of a takeaway. Do you have any guesses about why the current auto-interp pipeline fails? Ideas for how to improve it? It would be ideal if the paper had a bit more to say. What else can you say about what this result means for auto-interp or for interpretability more broadly?
>
> We think the current auto-interp pipeline probably fails for deep reasons. It is likely fundamentally impossible to translate the inner workings of a large language model into interpretable operations on independent, atomic features. This point has been argued by philosophers and cognitive scientists like Hubert Dreyfus for decades.
>
> On the other hand, incremental improvements may be possible with concerted work in two directions. One direction is that of improving the reconstruction error of sparse coders with architectural improvements. The other is to use more compute for generating explanations. Generating more explanations per feature, or using iterative methods to generate explanations might get us further improvements.
>
> > Do you think that your overall negative result is more likely to reflect the limitations of auto-interp or the limitations of SAEs themselves? I wonder whether, if we had stronger LLMs, which could, as agents, run experiments to test different hypotheses about latent explanations, or if we gave the simulator and explainer model other information (for instance about other internal latents in earlier network layers), whether something like this could eventually work?
>
> It is possible that, by making the explanations sufficiently long and detailed, we could eventually get significantly better results. Trivially, a very long explanation could perfectly capture the activation pattern of a latent by listing the weights of the base model and the transcoder, and describing how to compute the latent using these weights. But such an explanation would be useless. At the present time we can’t precisely estimate how much of an improvement these approaches can bring, but we have not had too much success with this approach, although it is possible that that might be due to the extreme costs necessary to explain latents in such a way.

---

> > ### Comment · Reviewer_q1xE · 2025-08-03
> >
> > Thanks for your comments. I think these sorts of conversations about whether certain kinds of interpretability are possible in principle are very interesting and worthwhile, though they are broader than the scope of any one paper. I expect I'll keep my score the same at a borderline accept.

---

### Official Review · Reviewer_6nTt · 2025-07-02

**Clarity:** 2
**Significance:** 2
**Originality:** 2
**Rating:** 3
**Confidence:** 1

**Summary:**

The paper examines whether we can rewrite an LLM using sparse coding (sparse auto encoder, transcoder). The paper proposes a pipeline using an LLM to find patterns in the target LLM's activations. They then propose to change the activations to control the output of the target LLM. The paper concludes that "we can't"

**Questions:**

Please see above

**Ethical Concerns:**

["NO or VERY MINOR ethics concerns only"]

**Final Justification:**

I raised my score because the response provides some clarification. However, in general, I believe the paper can be improved further
1. The writing in total is still difficult to read. The author response does make some clarification, but given the wording of the current submission, I don't think it's good enough for a general audience.
2. I still believe that the over-generalization problem of the paper should be addressed more carefully, probably extending the authors' arguments with references.
Given that the submission doesn't reach the page limitation, I believe it has lots of potential.

**Limitations:**

Yes

**Paper Formatting Concerns:**

No concerns

**Quality:**

2

**Strengths And Weaknesses:**

Although I think the idea of the paper is interesting, I found some drawbacks.

The paper quite difficult to read, partly because of some quite subjective statements, for instance, in the abstract "The greatest ambition of mechanistic interpretability is to completely rewrite deep neural networks in a format that is more amenable to human understanding, while preserving their behavior and performance". Is there a reference for that? I don't think the ultimiate purpose of interpretability is limited to NNs.

It's challenging to understand the logic of the paper. At the end of the introduction, the authors "puruse the ... idea", but then they state that "there are several roadblocks for this approach", yet "we see this test as a potential benchmark..." So why do the authors still believe in this approach?

In the abstract. Line 9-12, the paper says that "current sparse coding techniques and automated interpretability pipeline**s** are not up to the task". But the authors just try only ONE pipeline (proposed by them). How could they generalize that statement to other pipelines?

In the conclusion, the authors wrote "we proposed a new methodology for rigorously evaluating the faithfulness of natural language explanations...". I don't think anywhere in the paper shows any point about "faithfulness" in explanation.

In general, I can see that the paper present negative results using **their approach**, but I do think their claim is too wide in terms of generalization.

---

> ### Author Rebuttal · Authors · 2025-07-30
>
> > Summary:
> The paper examines whether we can rewrite an LLM using sparse coding (sparse auto encoder, transcoder). The paper proposes a pipeline using an LLM to find patterns in the target LLM's activations. They then propose to change the activations to control the output of the target LLM. The paper concludes that "we can't"
>
> We appreciate the reviewer's comments; however, we believe that this summary misinterprets the core objective of our research. While we do employ a Large Language Model (LLM) to identify specific activation patterns within a sparse coder, which itself is trained on the hidden representations of an LLM, the primary intent was not to manipulate these activations to control the output of a target LLM. Instead, our goal was to leverage the explanations provided by sparse coders as "concept bottlenecks" to reconstruct the original LLM's behavior.
>
> > Strengths And Weaknesses:
> Although I think the idea of the paper is interesting, I found some drawbacks.
> The paper quite difficult to read, partly because of some quite subjective statements, for instance, in the abstract "The greatest ambition of mechanistic interpretability is to completely rewrite deep neural networks in a format that is more amenable to human understanding, while preserving their behavior and performance". Is there a reference for that? I don't think the ultimiate purpose of interpretability is limited to NNs.
>
> Prominent researchers in mechanistic interpretability (MI) have consistently stated that the discipline's fundamental objective is to reverse-engineer the mechanisms of neural networks. For instance, the 2024 ICML workshop, accessible at https://icml2024mi.pages.dev/, explicitly says: “One emerging approach for understanding the internals of neural networks is mechanistic interpretability: reverse engineering the algorithms implemented by neural networks into human-understandable mechanisms.” Similarly, the most recent MI workshop at NeurIPS (https://mechinterpworkshop.com/) asserts: “The mechanistic interpretability field benefits from a rich diversity of approaches— (...) from reverse-engineering a model via bottom-up (...). But all are unified by the belief that there is meaning and structure to be found inside neural networks.”
>
> While “interpretability” can include research on machine learning models other than neural networks, the field of mechanistic interpretability is indeed entirely focused on neural nets. If there are other statements that the reviewer finds troublesome we would be happy to correct them.
>
> > It's challenging to understand the logic of the paper. At the end of the introduction, the authors "puruse the ... idea", but then they state that "there are several roadblocks for this approach", yet "we see this test as a potential benchmark..." So why do the authors still believe in this approach?
>
> This paper starts with a hypothesis. Sparse coders were introduced as a way to break down how models represent things internally. Assuming they can be interpreted in a clear way, we should be able to rewrite a language model's inner workings using everyday language. We worked on presenting evidence against this hypothesis by performing several experiments. The results of these experiments are the roadblocks mentioned in the introduction, which makes us posit that one of our initial assumptions is wrong: either sparse autoencoders don't really break down internal representations well enough, or we can't understand them well enough yet. It's easy to check the first assumption by looking at how well they reconstruct the model activations or how much the model's performance drops when we add them to the computation graph, but the second point is harder to measure. Because of this, we suggest our method as a way to test how well we can understand sparse autoencoders. If sparse autoencoders were fully interpretable, it should be possible to at least partly rewrite a transformer using natural language.
>
> > In the abstract. Line 9-12, the paper says that "current sparse coding techniques and automated interpretability pipelines are not up to the task". But the authors just try only ONE pipeline (proposed by them). How could they generalize that statement to other pipelines?
>
> We agree with the reviewer that the language in this sentence might be a bit too general. We do use a single automated interpretability pipeline, but are confident that current automated interpretability pipelines all have the same problem, as they are based on the same idea for generating interpretations. We are not aware of any pipeline that would be able to generate explanations that would allow for a precision of 99%, without tanking the recall.
>
> > In the conclusion, the authors wrote "we proposed a new methodology for rigorously evaluating the faithfulness of natural language explanations...". I don't think anywhere in the paper shows any point about "faithfulness" in explanation.
>
> Faithfulness normally refers to how well explanations conform to the model’s behaviour. If our explanations don’t allow the simulator LLM to accurately predict the activations of the latent, we can say that these explanations are not faithful.
>
> > In general, I can see that the paper present negative results using their approach, but I do think their claim is too wide in terms of generalization.
>
> Throughout this paper, we have tried to balance claims of generality regarding our results with an awareness of their inherent limitations. It is true that our results are confined to the re-implementation of an MLP module using a transcoder. In this context, it is reasonable to conclude that the necessary precision of natural language explanations for this endeavor is not currently possible with current transcoder and automated interpretability technology.

---

> > ### Comment · Reviewer_6nTt · 2025-08-03
> >
> > I would like to thank the authors for the response, which clarify my several concerns. I increase my scores.
> > In general, I believe the paper can be improved further
> > 1. The writing in total is still difficult to read. The author response does make some clarification, but given the wording of the current submission, I don't think it's good enough for a general audience.
> > 2. I still believe that the over-generalization problem of the paper should be addressed more carefully, probably extending the authors' arguments with references.

---

### Note · Authors · 2025-08-12

In this work, we evaluated the interpretability of sparse autoencoders (SAEs) by attempting to re-write a language model using natural language explanations of SAE features. This is a very ambitious task and we tried to measure how far we are from being able to do it using a state of the art automated interpretability pipeline.

Some reviewers suggested that our claims are too general, but we are confident that current techniques cannot perform this task. Despite evaluating 7 different SAE architectures in SAEBench, Karvonen et al. found that the interpretability of the leading architectures— JumpReLU, (batch) TopK, and Matryoshka— were all very similar. The work of “Scaling Automatic Neuron Descriptions” from Choi et al. (2024) shows that finetuning the explainer model and inference time compute scaling are insufficient to achieve the degree of specificity that would be necessary to qualitatively change our results.

Another point that was raised by some reviewers is that we are not clear on what exactly is failing in the pipeline. In early experiments we found that LLMs are biased toward predicting a latent to be active, but even after correcting this bias with quantile normalization, they are still unable to solve this task adequately. Qualitatively, the explanations produced by LLMs tend to be too general. Even when they completely explain the contexts where the feature is active, they don’t explain the contexts where the feature is inactive. This explains why, without quantile normalization, the LLMs are biased toward predicting that a feature should be active. More specific explanations will necessarily be longer and more complex, however. There is a fundamental tradeoff between specificity and the simplicity of an explanation. In the limit, a book-length explanation might be able to completely explain the activation pattern of a feature, but this may not be very satisfying or useful in practice. Future work should explore the Pareto frontier of specificity and simplicity in SAE feature explanations.

---

### Decision · Program_Chairs · 2025-09-17

**Decision:**

Reject

**Comment:**

This submission is an effort to evaluate the hypothesis that sparse autoencoders can be used to reverse engineer a neural network into natural language. The basic idea is to train a sparse autoencoder, generate a set of natural language labels for the features, using the natural language labels to generate examplar inputs with a second LLM and recording the simulated feature values, then patching in the simulated feature values to see if the original LM performance degrades. The ideas is that translating from SAE features to natural language and then back to SAE features is a way of evaluating whether the features+natural language labels are effective for reverse engineering the network.

The strength of this submission is that this is a really creative idea that is executed well.

The weakness of the submission is that there are multiple failure points in the experimental pipeline, and so the negative result is confounded between the SAEs being low quality and the natural language labels for features being low quality. This is a real weakness of the approach, but I'm hesitant to reject an evaluation method for being unable to pinpoint the exact point of failure in the SAE pipeline. Recent works on steering and probing with SAEs suffer from a similar problem with the quality of the SAEs being confounded with the quality of the natural language labels for the SAEs.

Reviewer VTXa was particularly concerned about the role of a judge LM, but given that LMs are used to generate the labels for SAEs this doesn't seem a big issue to me.

Ultimately, this paper has borderline reviews and due to the many papers at NeurIPS we must reject the paper.